# Machine Learning Based Prediction of Squamous Cell Carcinoma in Ex Vivo Confocal Laser Scanning Microscopy

**DOI:** 10.3390/cancers13215522

**Published:** 2021-11-03

**Authors:** Cristel Ruini, Sophia Schlingmann, Žan Jonke, Pinar Avci, Víctor Padrón-Laso, Florian Neumeier, Istvan Koveshazi, Ikenna U. Ikeliani, Kathrin Patzer, Elena Kunrad, Benjamin Kendziora, Elke Sattler, Lars E. French, Daniela Hartmann

**Affiliations:** 1Department of Dermatology and Allergy, University Hospital, LMU Munich, 80337 Munich, Germany; sschlingmann@me.com (S.S.); Pinar.Avci@med.uni-muenchen.de (P.A.); kathrin.patzer.Extern@med.uni-muenchen.de (K.P.); elenakunrad@yahoo.de (E.K.); benjamin.kendziora@med.uni-muenchen.de (B.K.); elke.sattler@med.uni-muenchen.de (E.S.); lars.french@med.uni-muenchen.de (L.E.F.); daniela.hartmann@med.uni-muenchen.de (D.H.); 2PhD School in Clinical and Experimental Medicine, University of Modena and Reggio Emilia, 41125 Modena, Italy; 3Munich Innovation Labs GmbH, 80336 Munich, Germany; zj@munich-innovation.com (Ž.J.); vpl@munich-innovation.com (V.P.-L.); 4M3i Industry-in-Clinic-Platform GmbH, 80336 Munich, Germany; fn@m3i-muenchen.de (F.N.); ik@m3i-muenchen.de (I.K.); ikk@m3i-muenchen.de (I.U.I.); 5Dr. Phillip Frost Department of Dermatology & Cutaneous Surgery, Miller School of Medicine, University of Miami, Miami, FL 33136, USA

**Keywords:** squamous cell carcinoma, ex vivo confocal laser scanning microscopy, reflectance confocal microscopy, fluorescence confocal microscopy, digital pathology, digital staining, neural networks, convolutional networks, machine learning, Mohs surgery

## Abstract

**Simple Summary:**

Squamous cell carcinoma is the second most common type of skin cancer, with incidence rates rising each year. Micrographic urgery is the treatment of choice for large, aggressive, or recurrent lesions. To ensure complete removal, excised tissue is frozen or embedded in paraffin, cut by a microtome, and stained for examination by an expert Mohs surgeon or a dermatopathologist. Thus, resection of tumor is performed in multiple steps, resulting in delayed wound closure, patient discomfort, longer hospital stay, and in turn, higher healthcare costs. In the last few years, ex vivo confocal laser scanning microscopy (CLSM) has shown promising results in intraoperative almost-real-time detection of skin cancers. This technology is not yet widespread in part due to necessity of training for laboratory technicians, surgeon and dermatopathologists. To increase efficiency and objectivity in the image interpretation process, we have built a machine learning model to detect squamous cell carcinoma lesions in excised tissues.

**Abstract:**

Image classification with convolutional neural networks (CNN) offers an unprecedented opportunity to medical imaging. Regulatory agencies in the USA and Europe have already cleared numerous deep learning/machine learning based medical devices and algorithms. While the field of radiology is on the forefront of artificial intelligence (AI) revolution, conventional pathology, which commonly relies on examination of tissue samples on a glass slide, is falling behind in leveraging this technology. On the other hand, ex vivo confocal laser scanning microscopy (ex vivo CLSM), owing to its digital workflow features, has a high potential to benefit from integrating AI tools into the assessment and decision-making process. Aim of this work was to explore a preliminary application of CNN in digitally stained ex vivo CLSM images of cutaneous squamous cell carcinoma (cSCC) for automated detection of tumor tissue. Thirty-four freshly excised tissue samples were prospectively collected and examined immediately after resection. After the histologically confirmed ex vivo CLSM diagnosis, the tumor tissue was annotated for segmentation by experts, in order to train the MobileNet CNN. The model was then trained and evaluated using cross validation. The overall sensitivity and specificity of the deep neural network for detecting cSCC and tumor free areas on ex vivo CLSM slides compared to expert evaluation were 0.76 and 0.91, respectively. The area under the ROC curve was equal to 0.90 and the area under the precision-recall curve was 0.85. The results demonstrate a high potential of deep learning models to detect cSCC regions on digitally stained ex vivo CLSM slides and to distinguish them from tumor-free skin.

## 1. Introduction

Cutaneous squamous cell carcinoma (cSCC) is a subtype of keratinocyte cancer (KC), that usually presents as a solitary, firm papule, or plaque with a hyperkeratotic surface, on chronically sun exposed areas. Due to its high incidence [1] and risk of metastasis, it represents a major health concern [2]. Clinical diagnosis of cSCC may be challenging, due to overlapping clinical features with other skin neoplasms such as keratoacanthoma, basal cell carcinoma (BCC), Bowen’s carcinoma or Merkel cell carcinoma [3]. Therefore, a surgical excision [4] with a subsequent histopathologic evaluation play a critical role in accurate diagnosis, therapy and management [4].

Current workflow of the pathological examination relies on labor-intensive and time-consuming tissue processing procedures such as paraffin embedding and sectioning. In comparison to paraffin sections, frozen sections are less time-consuming and are therefore often preferred in situations where timely decisions are needed (e.g., Mohs surgery). Nevertheless, this rapid diagnostic process comes at the expense of partial loss in tissue cellular architecture caused by freezing artifacts, difficulty in cut resulting in tissue folds, and poor staining quality, just to name a few [5,6]. Furthermore, individual factors, such as pathologist’s level of experience and expertise may result in an inter-and intraobserver variability as well as variations in efficiency [7,8,9].

Ex vivo confocal laser scanning microscopy (ex vivo CLSM) is one of the diagnostic innovations developed to overcome current challenges in traditional pathology [10,11,12]. It allows instant, high-resolution, bedside imaging of intact freshly excised tissue samples at a subcellular level [13]. The integrated features such as optical sectioning and mosaicking enable visualization of the whole tissue samples (generally up to 25 × 25 mm) at various depths and up to a 550-fold magnification [14,15,16]. Techniques such as acetowhitening, and the possibility of fluorescent mode (FM) together with the use of contrast agents like acridine orange (AO) can reveal tissue structures or cell organelles that were not visible in reflectance mode (RM) [14,15,16]. In addition, the lately introduced digital staining (DS) software can effectively simulate the conventional hematoxylin and eosin (H&E) staining that most dermatopathologists are trained to interpret [17,18]. Ex vivo CLSM shall be distinguished from its in vivo counterpart, which is based on a single near-infrared laser source and is used in the daily practice for bedside, non-invasive and painless diagnosis of cutaneous malignancies and several skin diseases, from mite infestations to lupus [19,20,21]. Its images are displayed as grey scale and in horizontal section, so that their comparison to vertical histological sections requires a dedicated training [22,23].

A preliminary report by Longo and colleagues already described the feasibility of ex vivo CLSM for intraoperative diagnosis of cSCC. Among 47 mosaics, which corresponded to 13 tumor sections and 34 tumor margins, 43 could be evaluated and 41 agreed with the frozen histopathologic sections [24]. A subsequent study from our group further demonstrated that ex vivo CLSM using both RM and FM enables differentiation of in situ carcinoma from invasive SCC [25].

Ex vivo CLSM has also limitations. To date in fact, the majority of practicing physicians are novice readers of this emerging technology; the interpretation of acquired images requires dedicated training and experience, even for expert dermatopathologists who need to get used to the different imaging modi and to the DS [26]. In addition, inter- and intraobserver variability both by experienced and non-experienced examiners possesses a challenge [27]. Last but not least, detection of numerous features and diagnostic criteria described in the present literature may be confusing and lengthy, especially for a novice reader in an intraoperative setting where time is rather scarce [16,25,27,28].

Over the past decade, with the adoption of machine learning tools, medicine is entering the dawn of a new era. Technological advances in computer power, network speeds, big data storage systems, digital transformation in healthcare, as well as development of new advanced algorithms are accelerating the translation of artificial intelligence (AI) from bench to clinic. Medical fields, such as pathology and radiology, that rely on visual patterns and clues are already starting to adopt AI into their workflow [28,29,30]. The increasing interest in such technologies is also demonstrated by the fact that regulatory agencies in the USA and Europe have started focusing on solutions to improve the use of big data, to facilitate innovation and support public health [31,32]. This was also supported by the approval of several deep learning/machine learning based medical devices and algorithms in the last few years, which are mainly employed in radiology, cardiology, but also neurology, oncology, endocrinology, internal and emergency medicine and dermatology to support decision making, interpretation of medical images, early detection of subclinical pathologies, customize drug dosages, improve medication adherence and even to reduce waiting times [32,33].

To date several studies justified the urgent need for an accelerated transformation by demonstrating that computer assisted detection and diagnosis systems increases accuracy, while reducing time required for image interpretation [30,34,35,36]. However, in case of pathology, in part due to obstacles in implementation of a fully digital workflow and training requirements, adoption rate has been rather slow [37,38]. Provided that ex vivo CLSM images are acquired and stored digitally, integration of a machine learning algorithm is well suited for utilization of this technology. To date however, no systematic studies on the automated detection of cSCC on CLSM images have been reported. In an attempt to accomplish standardized, objective, and accurate image interpretation of cSCC images acquired by ex vivo CLSM, we tested the feasibility of integration of a machine learning algorithm into our image analysis process. We aimed to determine the diagnostic performance of a new MobileNet convolutional neural network (CNN) for the detection of cSCC and tumor-free areas on CLSM images.

## 2. Materials and Methods

### 2.1. Population

From September 2020 to April 2021, 34 freshly excised tissue samples from 29 patients were examined using the 4th generation ex vivo CLSM (Vivascope2500, MAVIG Germany GmbH, Munich, Germany). These included 22 invasive cSCC from 17 patients (5 females, 12 males, aged 61–99 years) and 12 tumor-free tissue samples of the same anatomical areas donated from the puckering skin occurring during surgical wound closures (“dog ears”) (12 patients). (Table 1) The inclusion criteria were either a histological diagnosis of a cSCC or a histological exclusion of any tumor tissue. The specimens were obtained from the Dermatosurgical Unit of the Department of Dermatology and Allergy, Ludwig Maximilian University (LMU) of Munich. Preoperatively, written informed consent was provided by each patient following the principles of the declaration of Helsinki. The study was approved by the local ethics committee of the LMU Munich (Protocol number 19-150).

### 2.2. Device and Staining Technique

The 4th generation ex vivo CLSM used in the current study (Vivascope2500, MAVIG Germany GmbH, Munich, Germany) combines two lasers of different wavelengths (488 nm and 638 nm), and allows the examiner to evaluate the samples in different modes up to 550-fold magnification: FM (488 nm), RM (638 nm) and overlay mode (OM). The device can also produce images that simulate the effect of H&E staining in the DS mode [39].

All fresh tissue samples were processed immediately after resection. They were stained in AO (0.04 mg/mL resp. 0.12 mmol/L, Sigma-Aldrich, St. Louis, MO, USA) for 30 s, subsequently washed in Dulbecco’s Phosphate Buffered Saline (PBS, pH 7.4) for 20 s and placed in 10% citric acid solution for 30 s. After this staining procedure, the samples were fixed on the object slide using a sponge pad and magnets, as described by Pérez-Anker et al. [40]. All samples were scanned vertically to enable exact correlation to the corresponding histological images. After ex vivo CLSM imaging, the samples were fixed in formalin, embedded in paraffin, and stained with H&E for conventional histopathological reporting.

### 2.3. Image Evaluation and Annotation Process

All histopathological slides were evaluated by the senior pathologist of the department. All confocal images were evaluated and segmented by a dermatosurgeon expert in imaging techniques and histopathology (DH), a dermatosurgeon expert in imaging techniques with histopathology knowledge (CR) and a trainee expert in imaging techniques with histopathology knowledge (SS) using the criteria defined by Hartmann et al. [25]. In case of discrepancy, the senior dermatopathologist was involved. Only invasive cSCC were considered for this study. After the histological and ex vivo CLSM diagnosis, the tumor tissue was manually annotated for segmentation by an expert dermatologist with experience in both ex vivo CLSM and traditional histopathology on the digitally stained ex vivo CLSM images using the program QuPath, as described by Bankhead, P. et al. [41], adapted after Arvaniti et al. [42]. All tumors were annotated and analyzed in DS mode.

### 2.4. Preprocessing Data and Creation of a Deep Neural Network

The original ex vivo CLSM images (roughly 10,000 × 10,000 pixels) were split into smaller mosaics of 256 × 256 pixels. Other mosaic sizes have not been tested. If more than 50% of the 256 × 256-pixel area was marked as tumor tissue, then the whole patch was considered as tumor and thus region of interest (ROI). Choosing a threshold of 50% generated a balanced representation of tumor tissue on the edges. The 22 invasive cSCCs generated in total 45,823 (39,423 tumor free and 6409 tumor) mosaics and the 12 tumor free images were split into 22,971 mosaics, which were each separately evaluated by the algorithm (Table 2).

Then a MobileNet [43], a light weight deep convolutional neural network known for using depth-wise separable convolutions, which decreases processing time (arXiv:1704.04861), was used. The report by Adam and Howard and colleagues demonstrated MobileNets to have a better performance in comparison to other popular models on ImageNet classification such as VGG 16 [43,44]. This model uses 3 × 3 depthwise separable convolutions, which equires 8 to 9 times less computation when compared to standard convolutions [43,45]. In this architecture, all layers are followed by a batchnorm [45] and rectified linear unit (ReLU) nonlinearity with the exception of the final fully connected layer, which has no nonlinearity and feeds into a softmax layer for classification [43,44]. After the convolutional blocks, we added a global average pooling layer to extrapolate the information of the convolutional layers into a single vector. Two neurons were connected, whose weights and bias were randomly initialized. For the convolutional blocks, fine-tuning of the ImageNet-learned parameters was performed, as described by Arvaniti et al. [42] to achieve superior results. The group postulated that the neural networks trained by randomly initializing the starting weights, show a tendency to overfit to the training set and become overly confident in their predictions, producing higher cross-entropy loss in case of misclassifications. On the contrary, fine-tuned networks that are additionally regularized by dropout, such as MobileNet, do not reach 100% accuracy in the training set and do not exhibit an explosion in the validation cross-entropy loss scores [42]. Of note, the random initialization of weights and bias on the 2 neurons should not have had a significant impact on our evaluation as it presents only 0.125% of the network (1026 parameters compared to 815,592 parameters in the convolutional layers).

Adam optimizer [46] was applied to adjust the weights during the training. The learning rate was 1 × 10^−5^. The model was trained to minimize cross-entropy loss [47]. We used a dropout of 0.2 to avoid overfitting. To further combat overfitting, additional preprocessing steps were used on the training data, e.g., rotation, zooming and flipping. One training session took roughly 4 h on a NVIDIA GeForce GTX 1080 ti.

During the evaluation phase, the trained MobileNet was applied to the entire ex vivo CLSM image in a sliding window fashion and generated pixel-level probability heatmaps for both classes (tumor and tumor-free). We used a hard threshold to extrapolate a segmentation mask from the heatmap, which corresponds to the output of the MobileNet. Analyzed mosaics were defined as ROI in case the algorithm predicted a probability higher than 0.3 as being tumorous, which we defined as the classification threshold. Before extrapolating the segmentation mask from the heatmap, Gaussian filtering of the heatmap was done, in order to enhance the visual representation of the heatmap, which had no impact on the performance. Further post processing steps were performed on the segmentation mask. Morphological operations of erosion and dilation were implemented, to remove possible isolated pixels and then restore the original segmentation mask.

### 2.5. Statistical Evaluation

Our sample size was not large enough to accurately estimate the model performance with a single partition and test data. Therefore, we have evaluated our model using 17-fold cross validation. After dividing the data set randomly into 17 groups, 16 served as training data, whereas the remaining one was used as a test data set for assessment. This procedure was repeated 17 times so each group could be used as a test data set once. Before statistical evaluation, a tissue separation mask was created to separate the background from the tissue. Otsu thresholding was applied for this purpose [48]. Sensitivity and specificity for detecting tumor skin were calculated for the entire dataset including all mosaics from the 22 invasive cSCCs as well as the 12 tumor-free images. The overall predictive value was analyzed by the area under the receiver operating characteristic curve (ROC) and under the precision-recall curve for the entire dataset. The area under the ROC determines whether the deep neural network is able to distinguish between true positives (cancer) and false positives in a non-casual pattern (>0.5 probability). The area under the precision-recall curve does not consider true negatives (true healthy skin) and was additionally used since we had unbalanced classes, analyzing the fraction of cancer-labeled tissue that is really cancer (precision) and the fraction of cancer tissue, which was labeled as such by the software (recall).

## 3. Results

To test the performance of a machine learning algorithm in detection of cSCC in images acquired by ex vivo CLSM, we collected fresh tissue scans of 22 invasive cSCCs from 17 patients (5 females and 12 males, mean age 78.6). Those were manually segmented for tumor areas and evaluated (Figure 1). Furthermore, CLSM images from 12 samples of tumor-free skin were included as a negative control. The most frequently involved body site was the head and neck area (including scalp, nose, cheeks, lips) (Table 1).

The overall sensitivity and specificity of the deep neural network in detecting cSCC and tumor free skin in the ex vivo CLSM images compared to the expert examination were 0.76 and 0.91, respectively, using a classification threshold of 0.3 along with aforementioned post processing steps. The threshold of 0.3 was used because it offered the best balance between sensitivity and specificity as it generates the point closest (measured with Euclidean distance) to (0.1) in the ROC curve. To obtain the ROC curve, we predicted heatmaps of the cases in the validation set of every fold in the cross validation (Figure 1). We grouped all the tissue into one set of pixels and calculated the specificity and sensitivity using 20 classification thresholds equally distributed between 0 and 1. The precision-recall curve was calculated in the same manner. The area under the ROC curve was equal to 0.90, the area under the precision-recall curve was 0.85 (Figure 2).

By separately analyzing cSCC group and tumor free skin group, we reached a sensitivity of 0.76 and a specificity of 0.88 for cSCC and a specificity of 0.92 for tumor free skin (sensitivity was not calculated as it was not applicable in this case) (Table 3).

Following causes of false positives were observed: sun-damaged skin of elderly patients, dense inflammatory infiltrates surrounding the tumor masses, sebaceous glands and hair follicles. In one case of the negative control group (tumor free skin), clear cut sebaceous glands were misclassified as tumor tissue (Figure 3).

## 4. Discussion

The surgical management of cSCC, especially in head and neck area, is often performed in multiple steps with delayed wound closure, resulting into higher healthcare costs and patient discomfort. To accelerate such time consuming diagnostic process, Mohs micrographic surgery on frozen sections is used in selected centers worldwide, but its efficacy in high-risk tumors still needs further investigation [49].

Ex vivo CLSM has opened new horizons in bedside histology since it allows rapid analysis of freshly excised tissue after a short staining time with AO. When necessary, the identical samples can be processed for conventional dermatopathology, by H&E staining or immunohistochemistry, with no harm for the tissue [11,50]. The time required for the ex vivo CLSM staining, and image acquisition usually consists of few minutes depending on the sample size. The images are acquired in all CSLM modi simultaneously, and instantly stored. This provides significant advantages since slides can be analyzed intraoperatively. Further, several reports suggest that digitalization of pathology can potentially increase the efficiency of the slide reading process and reduce the related costs [51]. Ex vivo CLSM is already being used routinely for margin mapping of BCC in selected centres worldwide, providing a valid alternative to conventional Mohs surgery. Studies demonstrating the potential use of this modality in detection of other tumor types such as melanoma, cSCC [25,39,52,53], prostate and breast cancer [26,54,55,56] are already available in the literature. For instance, different types of prostatic and periprostatic tissue could be successfully differentiated on ex vivo CLSM images [26]. In addition, potential applications of ex vivo CLSM are not limited to the field of cancer but can also be expanded to characterization of autoimmune and inflammatory skin diseases such as pemphigus, pemphigoid, lichen planus, lupus erythematosus and vasculitis [13,57,58,59,60,61].

In order to further optimize the diagnostic process, deep learning models could assist the pathologists and Mohs surgeons to rapidly and precisely detect tumor ROI [62,63]. In the field of conventional pathology, various applications incorporating machine learning algorithms have generally shown a satisfactory performance to detect cell nuclei [64], mitosis [65], glands [8] and blood vessels [66]. Moreover, machine-learning algorithms have the capability to extract subtle features in acquired images beyond what is perceived by the human eye [67], and in turn identify patterns and associations [68,69,70].

Although pathology represent an ideal field of application for AI, its routine use has been held back by technical limitations, such as the amount of time required for slide digitalisation. Since CLSM automatically acquires and stores digital images, it stands out as an optimal candidate for the development of AI-driven diagnostic workflows. Preliminary studies have reported promising results, for instance in the enhanced automated detection of BCC [71]. Combalia et al. recently described a machine-learning assisted ex vivo CLSM pathology model, which was able to diagnose BCC with a sensitivity and specificity of 88% and 91%, respectively [71]. The model performance was even more satisfying when compared to analogous studies on digitally scanned H&E stained traditional pathology slides [62,63].

Our study experimented the utilization of a light weight deep CNN, which uses depth-wise separable convolutions for the automated detection of cSCC tissue on digitally stained ex vivo CLSM images. This trained algorithm was used to produce automated tumor positivity maps, in which the cSCC regions were marked and color-coded for display. Our model performed well by demonstrating an overall sensitivity of 0.76 and overall specificity of 0.91 for distinguishing between cSCC and tumor-free regions.

Such preliminary data shed light and hope to the potential benefits of deep learning algorithms in skin pathology since they can significantly accelerate the workflow via automated detection of tumor ROI directly on digitally stained ex vivo CLSM images. Especially novice examiners can highly benefit from such detection algorithms as they can focus on diagnostically relevant parts of the image in less time and get help from the algorithm. This improves not only the diagnosis of cSCC, but also the completeness of its resection by marking the tumor margins.

The innovative AI driven pathology in the ex vivo CLSM brings a potential improvement into the Mohs surgery workflow, as well as possible reduction of the patients’ burden in terms of time spent in the operating theatre, number of local anesthesia procedures, pain, and other surgery related complications. This procedure is not only alluring in surgical resection of malignant cutaneous tumors but can be potentially applied to other types of tumor surgeries aiming to obtain free margins before a closure. Algorithms able to correctly distinguish between precancerous and cancerous lesions could play a significant role in tumor mapping in the field of skin or prostate cancer. AI could also be used for developing intelligent imaging training systems for pathologists and physicians. A similar concept could be applied to in vivo CLSM images; even if their segmentation might be harder to perform due to the horizontal orientation of the images. Considering that the interpretation of in vivo CLSM images require a high level of experience and substantial training, the benefits of AI driven algorithms might be huge. Potential application fields comprehend AI driven image interpretation at bedside for tumor recognition, but also training programs for physicians and pathologists and support to decision making in the context of telemedicine.

It is of importance to note that this study had several limitations. The pilot algorithm is limited by the small sample size. A more conspicuous data is necessary to achieve higher area under the curve and to mimic real-world scenario. Moreover, our model was based on a binary distribution, only including two categories: invasive cSCC and tumor-free skin. This excluded a priori early stages of cSCC such as actinic keratoses and Bowen’s disease, which both display a certain grade of cytoarchitectural dysplasia. Our algorithm tended to mark certain regions of elderly sun-damaged skin as ROI. Dense inflammatory infiltrates surrounding the tumor masses, sebaceous glands and hair follicles could be confounded with the tumor itself because of a seemingly alike morphology (Figure 3). Analogous issues were observed in other studies conducted on digitally scanned conventional H&E stained BCC slides [63]. Similar false positive results also do occur during assessment of CLSM images by experienced dermatohistopathologists, as these structures are difficult to differentiate [72]. Therefore, this limitation is not confined to the algorithm itself and partly stem from the reduced contrast and image quality. Rather than a binary distribution, introducing for example the feature of peritumoral inflammation as an independent category could probably increase the specificity of the algorithm

In regions where image texture of tissue was similar to that of the background, the deep learning model was not always able to distinguish between the tissue and the background. Due to irregular tissue surface, out-of-focus areas led in few cases to a reduced image quality and suboptimal contrast. A clear example of this crucial issue is represented by repeated exclusion of tumor free regions displayed as blurred, and misinterpreted by the algorithm as background. Novel AI approaches such as feature engineering can address this problem by automatically identifying out-of-focus regions and subsequently adding extra set of focus points to these areas [29]. Another algorithm pitfall happened in few cases of squamous eddies that were recognized as background due to similarity in image texture. Subtracting the background prior to training and an improved background subtraction method which integrates a deep learning algorithm may overcome these current limitations.Moreover, despeckling neural networks and stack collapsing could further optimize image quality [73]; the latter combines in fact areas of highest contrast from various optical slides acquired at different focal Z-planes, reducing false negative results arising from air artefacts [71].

## 5. Conclusions

To sum up, we demonstrated in this proof-of-principle study that deep learning models can detect cSCC regions on digitally stained ex vivo CLSM images and distinguish them from tumor-free areas with a good sensitivity and specificity.

This might lay the foundations for an improved micrographic surgery workflow in which AO-stained fresh tissue is scanned by ex vivo CLSM, digitally stained and automatically analyzed for cSCC. The new workflow might be more efficient and cost-effective than the currently established standard of procedures. It might also be applied when the dermatopathologist is not available on site, through a teledermatology consultation platform.

Our results should serve as a hint for developing new standardized deep neural algorithms for the automated detection of tumor tissue on ex vivo CLSM images.

Further studies with larger multi-institutional studies are warranted to develop standardized acquisition methods and to increase the sensitivity and specificity of deep neural networks for detecting cSCC and to improve the distinction between different stages of KC.

## Figures and Tables

**Figure 1 cancers-13-05522-f001:**
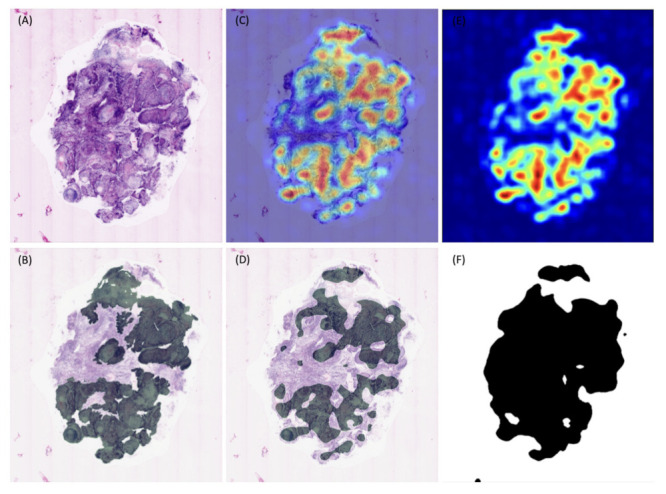
Image processing beginning from the digital staining mode of the ex vivo confocal image (**A**) as the ground truth; to the segmentation step performed by an expert, highlighted in green (**B**), heatmap overlaid on the tissue, the colormap used is jet, which ranges from blue to red, where red signifies tumor tissue and blue signifies tumor free tissue (**C**), as well as tissue marked as ROI by the algorithm (**D**), heatmap (**E**) and the tissue separation mask (**F**).

**Figure 2 cancers-13-05522-f002:**
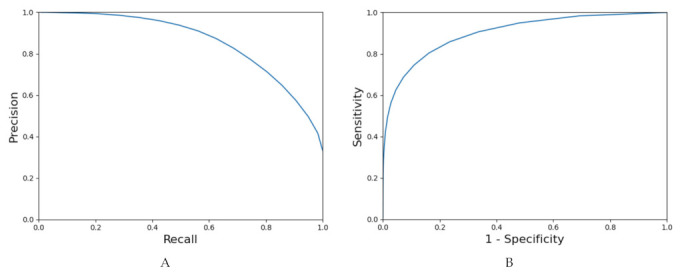
Precision-recall (**A**) and ROC curves (**B**).

**Figure 3 cancers-13-05522-f003:**
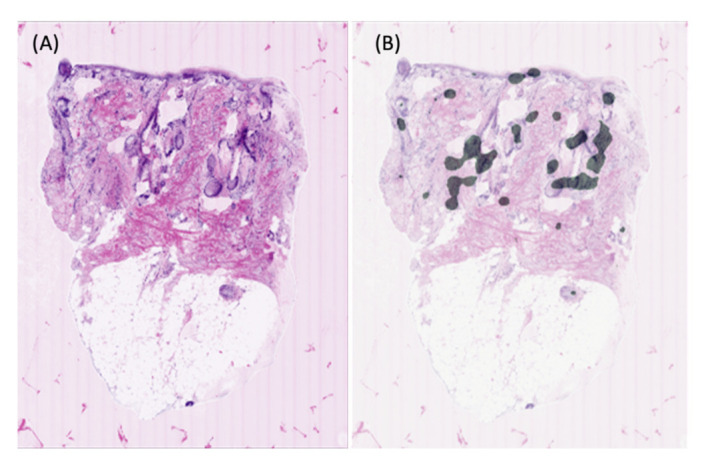
One of the pitfalls of the algorithm; tendency to misclassify sebaceous glands as tumor tissue. (**A**) Ground truth (tumor-free tissue with sebaceous glands). (**B**) Segmented areas by the algorithm considered as tumor.

**Table 1 cancers-13-05522-t001:** Demographic data of the cSCC patients included in the study (17 patients with 22 tumors), y = years.

Epidemiological Data		
Age	Average (y)	78.6
Gender	Female (%)	29.4
	Male (%)	70.6

**Table 2 cancers-13-05522-t002:** Number of mosaics considered ‘tumor-free’ and ‘carcinogenic’.

Group/Number of Mosaics	Tumor-Free Mosaics	Tumor Mosaics
Cutaneous squamous cell carcinoma (*n* = 22)	39,423	6409
Tumor-free skin (*n* = 12)	22,971	0
Total	62,394	6409

**Table 3 cancers-13-05522-t003:** Sensitivity, Specificity, Area under the precision-recall curve, area under the ROC curve.

Metric	Value
Sensitivity	0.76
Specificity	0.91
Area under ROC curve	0.90
Area under precision-recall curve	0.85

## Data Availability

Fully anonymised data are available on request; The software details are intended to be protected by a patent.

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
