# Peer review of "Machine Learning Based Prediction of Squamous Cell Carcinoma in Ex Vivo Confocal Laser Scanning Microscopy"

_cancers, 2021, doi:10.3390/cancers13215522_

Round 1

Reviewer 1 Report

Comments and Suggestions for Authors

I find the paper suitable for publications with a few minor adjustments:

Methods:

-       Can you please better describe who evaluated the FCM SCC images, who evaluated the histological slides and who performed the segmentation on the FCM images?

-       I appreciate your precision in describing the technical and statistical parts; however, I think this section would benefit of some cuttings because of repetitions. For example: 

we have assigned 20 equal interval treshold values set between 0 and 1 to use them as classification treshold to transform the heatmap (Fig. 1E) into a segmentation mask (Fig. 1D)”  in paragraph 2.4 and “We chose 20 different thresholds valued from 0 to 1, where each converted the heatmap into a segmentation mask by classifying every probability above the threshold into 1 (tumor), otherwise it was classified as 0 (not tumor)” in paragraph 2.5.

Figures and Tables:

-       Please add one more table including sensitivity and specificity values, eventually additionally the area under the curve in numbers; I find it more intuitive to read.

-       Please correct the typo sensitivity in the ROC curve

Discussion:

-       I think a couple of additional sentences focused on the advantages of FCM in daily life might enrich the second section of the discussion.

-       Analogously, I suggest adding a short paragraph about potential future applications of AI in FCM.

Author Response

see attachment please

Reviewer 2 Report

The paper target an important challenge, i.e., the need for fast and accurate machine vision tools to facilitate cancer diagnostics using laser scanning microscopy techniques. In this sense the work is important. What is problematic is that the proposed methodology is insufficiently described, and the results are vaguely presented. Thus the paper needs to be improved with respect to these aspects before being publishable.

Major concerns:

  • The dominant algorithm used is referred to as MobileNet; however, nowhere in the manuscript is the algorithm described, neither references provided. Sufficient information should be given in order for other labs to be able to reproduce the data. This needs to be corrected for.
  • The results section is insufficient. In principle the paper only contains one figure (Figure 2) with results, as two single-line curves. (Tables 1 and 2 describes demography and description of data, and are therefore not part of results.) These ROC curves in this only figure are difficult to interpret. What do these curves say? It can be assumed that the curves will vary depending on what parameters the network is trained/specified. Therefore, the results section would have been more informative if different ROC curves were to be presented for different parameter settings of the network. Then it would actually convey some results with respect to how the network is optimized in order to attain the best conditions for the discrimination. Furthermore, Figure 1 (presently only discussed in the methods section) should preferably be elaborated on and moved to the results section since this is the figure where actually images and discrimination are presented.

Additional comments:

  • It is somewhat unclear, if the motif behind the proposed technology is to develop a method that ultimately can be translated to a bedside in vivo approach, or if the target application is to replace conventional histopathology in Mohs surgery. Considering that acridin orange is applied as a fluorophore, it is assume that the latter ex vivo application is the target, but this should preferably be clarified. If this is the case, the discussion related to the advantage of ex vivo CLSM compared to digitized histopathology should be expanded, e.g. the optical rather than mechanical sectioning. Furthermore, it would be interesting if a discussion about how the proposed technology potentially can be translated to in vivo conditions can be included.
  • In the abstract it is mentioned that regulatory agencies have approved numerous deep learning protocols for medical devices. If this information is to be included, references should be included and discussed in the manuscript background/introduction. How does the proposed approach related to these approved technologies?
  • Section 2.4 contains a lot of terminology that is not sufficiently described, e.g. Adam optimizer, learning rate, cross-entropy, etc… How is classification threshold defined? The meaning of the heatmap is unclear. What parameter is the colormap representing?
  • Only Figure 1E and 1D is referred to in the paper. Where is the rest of Figure 1 described/discussed?
  • The error analysis introduced in the discussion, should preferably be reported as results. Same thing as for Figure 3.

Author Response

see attachment please
